# The Japanese version of the Generalized Problematic Internet Use Scale 2 (GPIUS2): Psychometric evaluation and analysis of the theoretical model

**Sayaka Yoshimura** [1,2,3]*, **Mami Shibata**[4], **Yusuke Kyuragi**[4], **Kei Kobayashi**[4], **Morio Aki**[4], **Toshiya Murai**[4], **Hironobu Fujiwara** [4,5,6]

1 Faculty of Human Health Science, Graduate School of Medicine, Kyoto University, Kyoto, Japan, 2 Organization for the Promotion of Neurodevelopmental Disorder Research, Kyoto, Japan, 3 Department of Developmental Disorders, National Institute of Mental Health, National Center of Neurology and Psychiatry, Kodaira, Japan, 4 Department of Neuropsychiatry, Graduate School of Medicine, Kyoto University, Kyoto, Japan, 5 Artificial Intelligence Ethics and Society Team, RIKEN Center for Advanced Intelligence Project, Tokyo, Japan, 6 The General Research Division, Osaka University Research Center on Ethical, Legal and Social Issues, Kyoto, Japan

* yoshimurasayaka@gmail.com

**Data Availability Statement:** Data cannot be shared publicly for ethical reasons. This project has

## Abstract

### Background

The Generalized Problematic Internet Use Scale 2 (GPIUS2) is a self-administered questionnaire that evaluates problematic internet use (PIU) from a multidimensional perspective. We analysed the psychometric properties and adequacy of the theoretical model of Japanese version of the GPIUS2.

### Methods

This study included 291 healthy Japanese adults (median age = 25 years; interquartile range 22–43 years; 128 women) who completed the GPIUS2 and several other questionnaires evaluating the degree of PIU, self-esteem, depression, and impulsivity.

### Results

Exploratory factor analysis (EFA) revealed a similar factor structure between the original and Japanese versions of the GPIUS2, with only minor differences in item composition. Higher-order confirmatory factor analyses revealed a good overall fit for the factorial model suggested by EFA, indicating adequate construct validity. The model showed acceptable internal consistency. Partial correlation analyses between GPIUS2 and other measures, with age as a control variable, revealed good convergent validity. Finally, structural equation modelling showed a good fit to the data, supporting the cognitive-behavioural model of Caplan (2010).

not been granted ethics to release the data publicly and we did not obtain the consent of participants to provide the datasets to third parties. However, researchers who meet the criteria for access to confidential data can contact the Ethics Committee of Kyoto University Graduate School and Faculty of Medicine at: ethcom@kuhp.kyoto-u.ac.jp; or the corresponding author at: yoshimura. sayaka.3x@kyoto-u.ac.jp.

**Funding:** Funding: SY received a Grant-in-Aid for Young Scientists (Japan Society for The Promotion of Science, 20K13821). TM received support from a Grant-in-Aid for Scientific Research (A) (Japan Society for The Promotion of Science, 19H00518), Gant-in-Aid for Transformative Research Areas (A) (Japan Society for The Promotion of Science, JP21H05173), Grant-in-Aid for Scientific Research (B) (Japan Society for The Promotion of Science, 21H02849), and Strategic International Brain Science Research Promotion Program (Brain/ MINDS Beyond) (21dm0307102h0003) of the Japan Agency for Medical Research and Development (AMED). HF got a Grant-in-Aid from the Smoking Research Foundation. The funders had no role in study design, data collection and analysis, decision to publish, or manuscript preparation.

**Competing interests:** The authors have declared that no competing interests exist.

## Conclusions

The Japanese version of the GPIUS2 has good psychometric properties and the theoretical model of the original GPIUS2 is applicable to Japanese adults.

## Introduction

The internet has become an essential tool for daily life, and has revolutionized how people acquire information and communicate with each other. However, a substantial proportion of people exhibit problematic internet use (PIU), which negatively affects daily functioning; Impact of internet use on daily life, such as work and school achievement, is now considered more important for determining whether internet use is problematic than the time spent on the internet, and the online activities engaged in (e.g., [1]). Over the past two decades, research has revealed that PIU has negative effects on mental health [2] and often coexists with psychiatric disorders such as depression, obsessive-compulsive disorder, and neurodevelopmental disorders ([3]; for reviews, see [4–7]). Furthermore, PIU shares a common neural basis with other addictive behaviours, such as binge eating [8].

According to Davis [9], who provided one of the most influential theories of internet use, PIU comprises cognitive and behavioural elements including social, psychopathological, and physiological functioning. Davis proposed that PIU can be divided into two types: specific and generalized. Specific PIU refers to a particular type of internet use (e.g., online gambling), whereas generalized PIU includes more general behaviours (e.g., socialising via the internet) than specific PIU. In addition, the model suggests that cognition plays a more critical role in generalized than specific PIU.

Most tools available to assess PIU, such as the Internet Addiction Test (IAT [10]) and Problematic Internet Usage Questionnaire [11], provide evidence of its multidimensional nature. However, their use of a one-dimensional structure and a theoretical approach have prevented an in-depth understanding of PIU [12].

The second version of the Generalized Problematic Internet Use Scale (GPIUS2 [13]) was one of the first scales developed to evaluate generalized PIU from a multidimensional perspective. The GPIUS2 consists of four subscales: preference for online social interaction (POSI), mood regulation, deficient self-regulation (further divided into cognitive preoccupation and compulsive internet use), and negative outcomes. POSI evaluates the preference for online over offline social communication. Mood regulation assesses the tendency to regulate mood through internet use. Deficient self-regulation involves difficulty in appropriately monitoring and judging the pattern of the individual's internet use, and adjusting internet-related behaviors. Cognitive preoccupation includes obsessive thoughts concerning the internet, and compulsive internet use refers to failure to control internet use. Negative outcomes refer to adverse effects of internet use. These subscales were based on Davis' cognitive-behavioural model [9], Caplan's work on POSI [14, 15], and the sociocognitive model of dysregulated internet use of LaRose and colleagues [16, 17].

The GPIUS2 model hypothesized that POSI and mood regulation increase deficient self-regulation, resulting in negative outcomes (Fig 1). Adequate construct, discriminant, and convergent validity, as well as good internal consistency, have been reported for the original GPIUS2 and several translations thereof [12, 18–22]. The results of previous studies empirically support the theoretical model of the GPIUS2, which allows for the evaluation of multiple

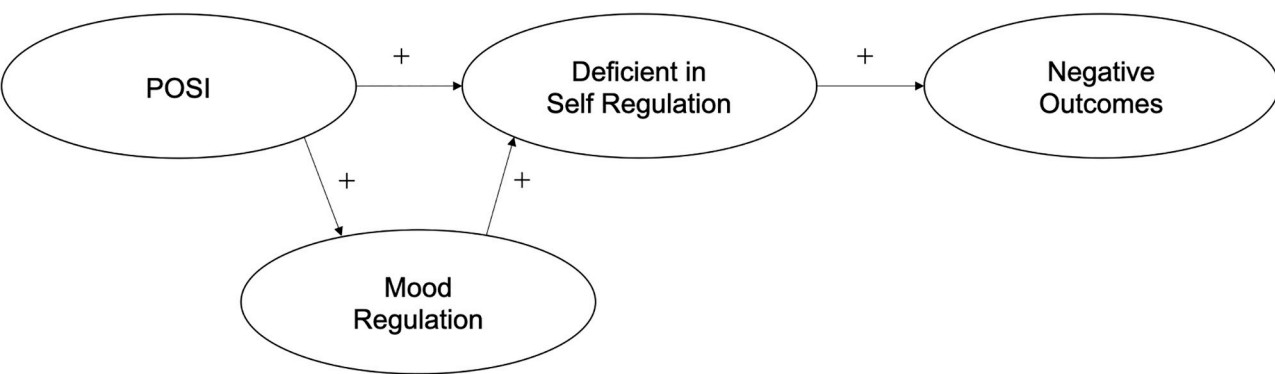

**Fig 1. Cognitive-behavioural model of generalized problematic internet use (Caplan, 2010 [18]).**

aspects of generalized PIU. GPIUS2 scores are correlated with many psychological and health indexes, indicating that it is a valuable clinical and research tool for assessing generalized PIU.

The primary purpose of the present study was to translate the GPIUS2 into Japanese and evaluate the psychometric properties thereof. The secondary purpose was to assess the adequacy of the theoretical model of generalized PIU (Fig 1) when applied to Japanese adults. We hypothesised that the cognitive-behavioral model of generalized PIU would be applicable to Japanese adults, i.e., that POSI is a direct positive predictor of mood regulation (H1) and deficient self-regulation (H2); mood regulation is a direct positive predictor of deficient self-regulation (H3); Deficient self-regulation is a direct positive predictor of negative outcomes (H4); mood regulation mediates the relationship between POSI and deficient self-regulation (H5); deficient self-regulation mediates the relationship between POSI and negative outcomes (H6); and deficient self-regulation mediates the relationship between mood regulation and negative outcomes (H7). We used the two-step modelling approach adopted in previous GPIUS2 validation studies [12, 18, 21, 22], as described in the Methods.

## Materials and methods

### Participants

This study included 291 Japanese adults (128 women and 163 men; median age = 25 years; interquartile range 22–43 years; age range = 18–78 years) participating in two projects targeting adults aged ≥18 years (one evaluating the relationship between lifestyle habits and mental health, and an imaging study on schizophrenia). The participants had no psychiatric or severe neurological conditions, and served as controls for comparison with psychiatric patients. Two experienced psychiatrists conducted interviews to screen for psychiatric and severe neurological diseases. The participants provided written informed consent for use of their information when they participated in the original project.

The study was approved by Kyoto University Graduate School and Faculty of Medicine, Ethics Committee, and performed in accordance with the 1964 Declaration of Helsinki and its later amendments.

### Measures

**GPIUS2.** The GPIUS2 is a 15-item self-report questionnaire that evaluates generalized PIU. The 15 items are scored on an 8-point Likert scale (1 = strongly disagree, 8 = strongly

agree). The total GPIUS score ranges from 15 to 120, with higher scores indicating more severe generalized PIU.

The original version of the GPIUS2 [18] was translated into Japanese. A psychiatrist experienced in managing PIU-related cases and a cognitive science researcher, performed the translation. The researcher was proficient in Japanese and English and translated and adapted the psychological measures. Disagreements in wording were resolved through consultation with a specialist in psychometrics, to ensure functional and linguistic equivalence between the original and Japanese versions of the GPIUS2. Then, two bilingual couples, including a native English and native Japanese speaker, back-translated the draft. The cognitive science researcher and psychiatrist compared the back-translated and original versions to identify semantic inconsistencies; minor errors in Japanese were corrected.

The following four questionnaires were administered to evaluate the convergent validity of the GPIUS2.

**Internet Addiction Test (IAT).** The IAT [10] is a 20-item self-administered questionnaire that evaluates the degree of PIU from a behavioural perspective. This questionnaire has also been used to evaluate other PIU scales [19, 20, 22]. We administered the Japanese version of the IAT developed by the Division of Treatment of Internet Addiction and Research at The National Hospital Kurihama Alcoholism Center, and validated by Michie Hesselbrock (see https://kurihama.hosp.go.jp/hospital/screening/iat.html). The items are rated on a 5-point scale (1 = rarely, 5 = always). The total IAT score ranges from 20 to 100, with higher scores indicating more severe PIU. The scores for the Japanese version were classified as follows: 20–39, average internet use; 40–69, possible addiction; and 70–100, severe addiction.

In total, 134 participants (50 women; median age = 23 years; interquartile range 21–33.75 years) completed the IAT.

**Rosenberg Self-Esteem Scale (RSES).** Low self-esteem is considered to play a crucial role in generalized PIU development [9, 23]. The RSES [24] is a 10-item self-report questionnaire that evaluates self-esteem. The validity and reliability of the Japanese version of the RSES were confirmed by Uchida and Ueno [25]. The items in the Japanese version are rated on a 4-point scale (1 = strongly disagree, 4 = strongly agree). The total RSES score ranges from 10 to 40, with higher scores indicating higher self-esteem.

In total, 202 participants completed the RSES (80 women; median age = 24 years; interquartile range 21–40.75 years).

**Beck Depression Inventory-II (BDI-II).** The cognitive-behavioural model of Davis [9] proposes that the presence of psychiatric problems (e.g., depression) is necessary for generalized PIU. The BDI-II is a 21-item self-reported questionnaire that evaluates depression symptoms during the prior two weeks. Its validity was established by Kojima et al. [26]. The items are rated on a 4-point scale (0–3), with the total score ranging from 0 to 63 and higher scores indicating more severe depression symptoms. In the Japanese version, scores of 17–20 indicate borderline clinical depression.

In total, 282 participants (126 women; median age = 24 years; interquartile range 22–43 years) completed the BDI-II.

**Barratt Impulsiveness Scale version 11 (BIS-11).** PIU is conceptualized as an impulse control disorder [27]. The BIS-11 is a 30-item self-report questionnaire that evaluates impulsivity. The validity and reliability of the Japanese version of the BIS-11 were confirmed by Someya et al. [28]. The items on the Japanese version are rated on a 4-point scale (1 = rarely/never, 4 = almost always/always), with total scores ranging from 30 to 120 and higher scores indicating greater impulsivity.

In total, 137 participants (52 women; median age = 23 years; interquartile range 21–34 years) completed the BIS-11.

## Data analysis

First, we generated descriptive characteristics. The normality of the data was assessed using the Shapiro-Wilk test.

We used a two-step analysis procedure [29] to test the study hypotheses. This procedure was used to develop theoretical models and test their structure in previous GPIUS2 validation studies [12, 18, 21, 22]. First, we performed a higher-order confirmatory factor analysis (CFA) to validate the measurement model and assess its construct validity. Second, we performed structural equation modelling (SEM) of the causal relationships between constructs to validate model structure.

Jia and Jia [30] reviewed the analytical techniques used to develop PIU scales, and recommended CFA on several different models for assessment of construct validity. In addition, they recommended exploratory factor analysis (EFA) to extract latent factors. Thus, we conducted EFA of the GPIUS2 before the CFA. We used JASP (JASP Team, 2021) and AMOS to perform these analyses. JASP, an open-source statistical software package created by researchers at the University of Amsterdam, is implemented in several R packages.

After EFA, CFA was performed on four models (Fig 2). *Model 1* included a single factor (i.e., problematic internet use). *Model 2* included four factors (i.e., POSI, mood regulation, deficient self-regulation, and negative outcomes) without a higher-order factor. *Model 3* is Caplan's model, which provided the basis for the original GPIUS2 [18]. This model included five first-order factors, and a second-order factor of deficient self-regulation associated with two first-order factors (i.e., cognitive preoccupation and compulsive use). *Model 4* is the Japanese model, the factor structure of which was based on EFA. It differs slightly from *Model 3* in terms of the latent variable (i.e., item 12). We used root mean square error of approximation (RMSEA), standardized root mean square residual (SRMR), Bentler's comparative fit index (CFI), the Tucker–Lewis index (TLI), and the goodness-of-fit index (GFI) to identify the most valid model. RMSEA and SRMR values of 0.05–0.08 and < 0.05 indicate an acceptable and excellent fit, respectively. CFI, TLI, and GFI values > 0.90 and > 0.95 indicate acceptable and good fits, respectively. In the CFA, the parameters were estimated using the maximum likelihood method.

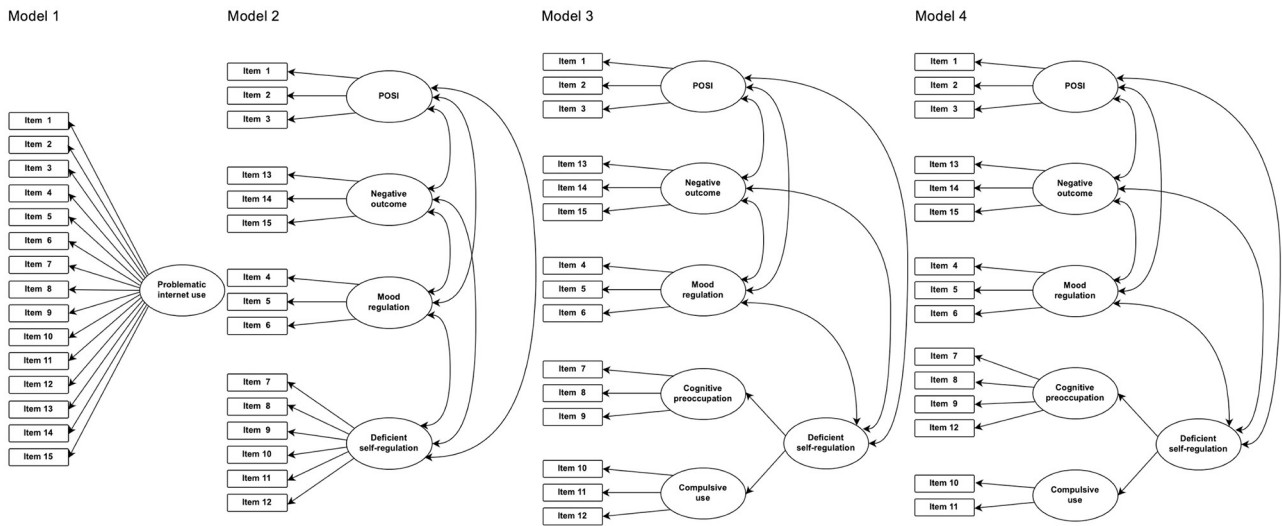

**Fig 2. The four models evaluated using confirmatory factor analysis (CFA).**

After confirming the most valid model by CFA, we computed Cronbach's α coefficients for the GPIUS2 subscales, and the full scale to evaluate internal consistency. We also calculated the correlation coefficients of the GPIUS2 subscales and full scale with multiple variables (i.e., IAT, RSES, BDI-II, and BIS-11) to evaluate the convergent validity. Although the number of participants who completed each questionnaire differed, all responses were included in the analysis.

Finally, in the second step of the analysis, we tested the causal relationships between constructs using SEM, to assess the adequacy of the cognitive-behavioural model of Caplan [18]. We assessed goodness of fit using the same measures applied for CFA.

## Results

### Descriptive statistics

The descriptive characteristics of the measures are presented in Table 1. All variables, except the RSES and BIS-11, were non-normally distributed. The IAT scores ($n$ = 134) revealed that 56.0% ($n$ = 75), 41.0% ($n$ = 55), and 3.0% ($n$ = 4) of the participants were normal, maladaptive, and problematic internet users, respectively.

### Psychometric properties of the GPIUS2

**EFA.** We performed EFA to investigate the factor structure of the Japanese version of the GPIUS2, using the maximum likelihood method in combination with oblimin rotation. Based on parallel analysis and visual inspection of the scree plot, we decided to use the same five-factor structure as the original GPIUS2, characterised by a lack of singularity (Bartlett's $\chi^2$ = 2384, $df$ = 105, $p$ < .001) and adequate sample size (Kaiser-Meyer-Olkin = .87). The five-factor solution included 15 items, accounted for 65.1% of the variance, and had robust factor loadings (0.52–0.95) (Table 2).

**Table 1. Descriptive statistics of the participants and the measures.**

| | $n$ | Woman (%) | Median | Interquartile range | Range | Shapiro-Wilk Statistic |
|---|---|---|---|---|---|---|
| Age (years) | 291 | | 25 | 22–43 | 18–78 | 0.845*** |
| Gender | | 128 (44.0%) | | | | |
| *GPIUS2* | 291 | 128 (44.0%) | | | | |
| POSI | | | 6 | 4–9.5 | 3–19 | 0.895*** |
| Mood regulation | | | 10 | 5–15 | 3–23 | 0.940*** |
| Deficient self-regulation | | | 16 | 10–23 | 6–44 | 0.947*** |
| Cognitive preoccupation | | | 9 | 5–13 | 4–30 | 0.905*** |
| Compulsive use | | | 7 | 4–10 | 2–16 | 0.934*** |
| Negative outcome | | | 4 | 3–7 | 3–21 | 0.767*** |
| Total GPIUS2 | | | 54 | 38–73.5 | 21–144 | 0.961*** |
| *IAT* | 134 | 50 (37.3%) | 39 | 31–47 | 20–79 | 0.960*** |
| *RSES* | 202 | 80 (39.6%) | 27 | 23–30 | 12–40 | 0.994 |
| *BDI-II* | 282 | 126 (44.7%) | 5 | 3–9.75 | 0–36 | 0.887*** |
| *BIS-11* | 137 | 52 (37.8%) | 62 | 56–70 | 41–95 | 0.982 |

GPIUS2: Generalized Problematic Internet Use Scale 2; POSI: preference for online social interaction; IAT: Internet Addiction Test; RSES: Rosenberg Self-Esteem Scale; BDI: Beck Depression Inventory; BIS: Barratt Impulsiveness Scale.

***$p$ < .001

**Table 2. Factor loadings of the GPIUS2 items in the final exploratory factor analysis.**

| | I | II | III | IV | V | $h^2$ |
|---|---|---|---|---|---|---|
| 1. I prefer online social interaction over face-to-face communication | | 0.95 | | | | 0.92 |
| 2. Online social interaction is more comfortable for me than face-to-face interaction | | 0.86 | | | | 0.73 |
| 3. I prefer communicating with people online rather than face-to- face | | 0.50 | | | | 0.38 |
| 4. I have used the Internet to talk with others when I was feeling isolated | | | 0.52 | | | 0.38 |
| 5. I have used the Internet to make myself feel better when I was down | | | 0.89 | | | 0.78 |
| 6. I have used the Internet to make myself feel better when I've felt upset | | | 0.76 | | | 0.68 |
| 7. When I haven't been online for some time, I become preoccupied with the thought of going online | 0.53 | | | | | 0.49 |
| 8. I would feel lost if I was unable to go online | 0.62 | | | | | 0.55 |
| 9. I think obsessively about going online when I am offline | 0.87 | | | | | 0.80 |
| 10. I have difficulty controlling the amount of time I spend online | | | | | 0.62 | 0.73 |
| 11. I find it difficult to control my Internet use | | | | | 0.86 | 0.76 |
| 12. When offline, I have a hard time trying to resist the urge to go online | 0.70 | | | | | 0.70 |
| 13. My internet use has made it difficult for me to manage my life | | | | 0.84 | | 0.71 |
| 14. I have missed social engagements or activities because of my Internet use | | | | 0.62 | | 0.48 |
| 15. My Internet use has created problems for me in my life | | | | 0.77 | | 0.70 |

The results also showed that the factor structure of the five-factor solutions was similar to that of the original GPIUS2. However, item 12, which loaded onto the "compulsive internet use" factor in the original version, was loaded onto Factor I, which was taken to be equivalent to the "cognitive preoccupation" factor in the original version. Considering that the contents of item 12 pertain to obsessive thoughts about internet use and inadequate behavioural control, we classified it as a "cognitive preoccupation" item. In accordance with the original version [18], we named Factors I–V cognitive preoccupation, POSI, mood regulation, negative outcomes, and compulsive internet use, respectively.

**CFA of the measurement model.** CFA was used to validate the measurement model. We performed CFA on the four models, as described in the Methods. The Japanese model (i.e., the model with first-order and second-order factors, and the latent variable of cognitive preoccupation affecting an item 12) showed the best fit to the data (Table 3), as indicated by robust factor loadings (range: 0.57–0.97, Fig 3). Fig 3 shows the results for CFA on the Japanese model.

**Internal consistency.** The Cronbach's α coefficients of the first-order factors were .79, .76, .75, .77, and .78 for POSI, mood regulation, cognitive preoccupation, compulsive use, and negative outcomes, respectively. The internal consistency of the second-order factor (i.e., deficient self-regulation) was $\alpha = .70$. These results showed that the internal consistency of the Japanese model was acceptable. The internal consistency of the full scale was $\alpha = .86$.

**Table 3. Goodness-of-fit of the models: Results of confirmatory factor analysis.**

| | $\chi^2$ | df | p | RMSEA | SRMR | CFI | TLI | GFI |
|---|---|---|---|---|---|---|---|---|
| *Model 1.* A single-factor model | 918.23 | 90 | < .001 | .178, 90% CI [.167, .188] | .113 | .646 | .586 | .692 |
| *Model 2.* Simple four-factor model | 271.61 | 84 | < .001 | .088, 90% CI [.076, .099] | .060 | .920 | .900 | .883 |
| *Model 3.* Caplan's original model | 239.41 | 82 | < .001 | .081, 90% CI [.069, .093] | .059 | .933 | .914 | .899 |
| *Model 4.* The Japanese model | 170.31 | 82 | < .001 | .061, 90% CI [.048, .074] | .055 | .962 | .952 | .926 |

RMSEA: root mean square error of approximation; SRMR: standardized root mean square residual; CFI: Bentler's comparative fit index; TLI: Tucker–Lewis index; GFI: goodness-of- fit index

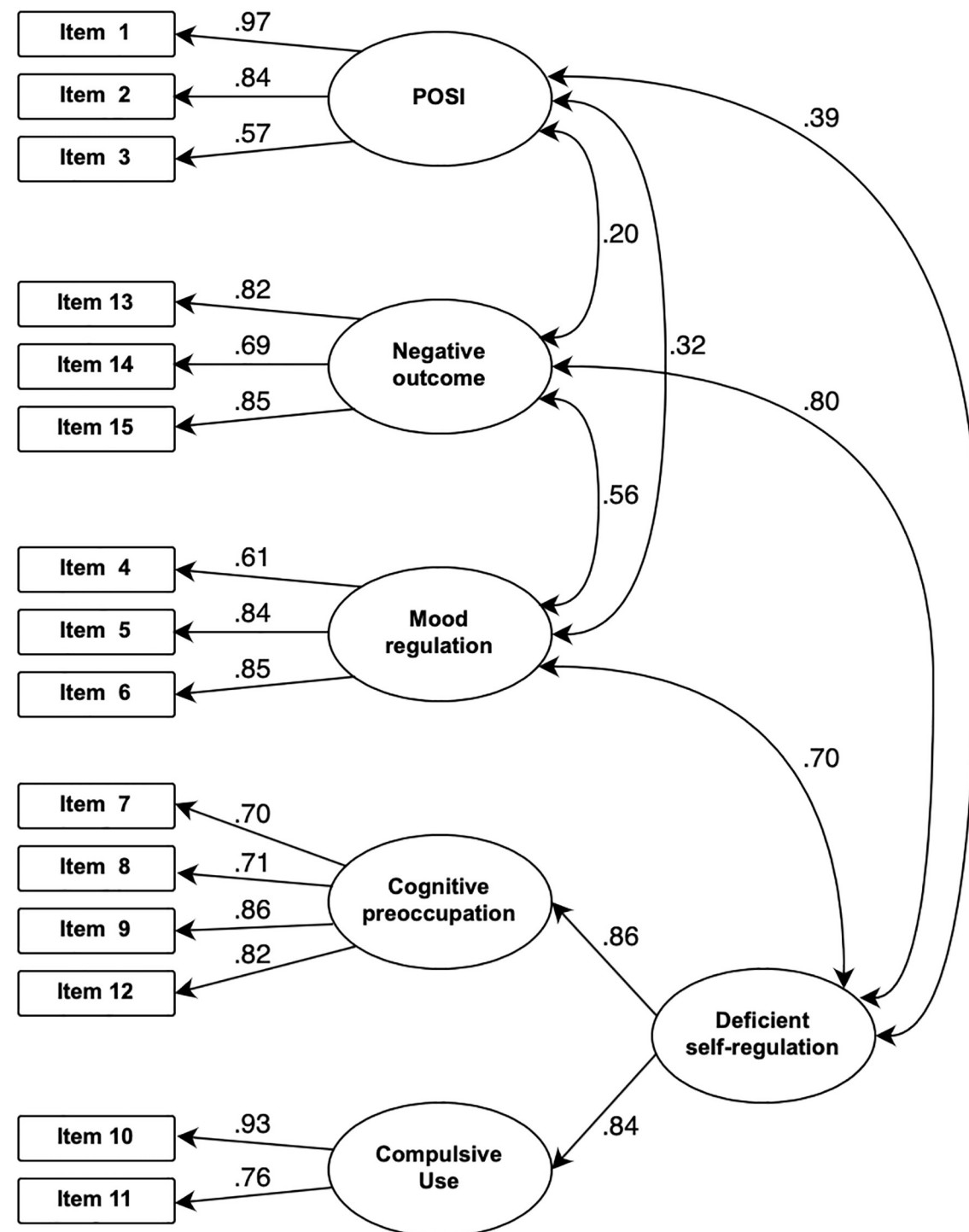

**Fig 3. Confirmatory factor analysis for the Japanese model of GPIUS2.**

**Convergent validity.** We assessed convergent validity by analysing the correlations of the GPIUS2 subscale and full-scale scores with related measures. The non-parametric Spearman's rank correlation was used because all measures, except the RSES and BIS-11, were non-normally distributed according to the Shapiro–Wilk test. There were age, but not gender,

**Table 4. Partial correlation coefficients between the GPIUS2 scales and IAT, self-esteem, depression, and impulsivity, with age as a covariate.**

| GPIUS2 subscales and GPIUS2 total | IAT (n = 134) | RSES (n = 201) | BDI-II (n = 282) | BIS-11 (n = 137) |
|---|---|---|---|---|
| POSI | .29*** | -.12 | .15* | .07 |
| Mood regulation | .58*** | -.21** | .23*** | .08 |
| Deficient self-regulation | .73*** | -.14* | .30*** | .17 |
| Cognitive Preoccupation | .57*** | -.11 | .23*** | .07 |
| Compulsive use | .74*** | -.14* | .30*** | .23** |
| Negative outcomes | .68*** | -.15* | .31*** | .14 |
| GPIUS total | .75*** | -.18* | .33*** | .16 |

GPIUS2: Generalized Problematic Internet Use Scale 2; POSI: preference for online social interaction; IAT: Internet Addiction Test; RSES: Rosenberg Self-Esteem Scale; BDI: Beck Depression Inventory; BIS: Barratt Impulsiveness Scale.

*$p < .05$;

**$p < .01$;

***$p < .001$

differences between the overall cohort (n = 291) and the subgroups that completed each questionnaire. We performed partial correlation analyses controlling for age, as age was correlated with all GPIUS2 subscale scores ($p < .002$ for all), with the exception of the POSI score.

Table 4 shows the significant partial correlation coefficients. IAT and BDI-II scores had significant positive relationships with GPIUS2 subscale scores, and there were significant negative relationships between RSES and GPIUS2 subscale scores. BIS-11 scores had significant positive relationships only with the compulsive use subscale scores. Similar results were observed after listwise deletion (*n* = 128, S1 Table).

**Analysis of the theoretical model.** After validating the measurement model, we tested the cognitive-behavioural model of Caplan ([18], Fig 1) using SEM (the second step of the two-step analysis procedure, [29]). Fig 4 shows the estimated standardized beta coefficients.

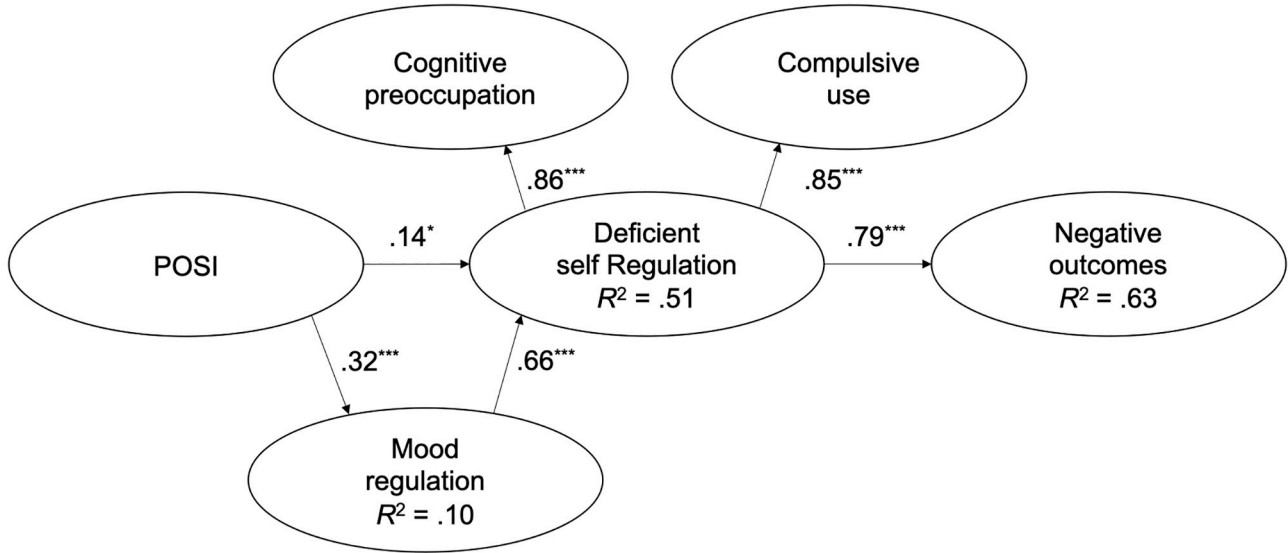

**Fig 4. Standardized coefficient estimates of the theoretical model.** *$p < .05$, ***$p < .001$.

The results revealed that the Japanese model fit the data well (Bartlett's $\chi^2$ = 175.20, $df$ = 85, $p$ < .001; RMSEA = .06 [90% confidence interval: .05–.07]; SRMR = .06; CFI = .961; TLI = .952; GFI = .923). Direct effect analysis showed positive relationships between factors. POSI was a significant positive predictor of mood regulation (standardized coefficient estimates; $\beta$ = .32, $p$ < .001) and deficient self-regulation ($\beta$ = .14, $p$ < .05). The use of the internet for mood regulation was a significant positive predictor of deficient self-regulation ($\beta$ = .66, $p$ < .001). In addition, deficient self-regulation was a positive predictor of negative outcomes ($\beta$ = .79, p < .001) (Fig 4). These results support the first four hypotheses described in the Introduction (H1–H4).

Mood regulation mediated the positive indirect relationship between POSI and deficient self-regulation ($\beta$ = .21, $p$ < .05). Deficient self-regulation mediated the positive indirect relationships between POSI and negative outcomes ($\beta$ = .27; $p$ < .05), and between mood regulation and negative outcomes ($\beta$ = .52; $p$ < .05). The variables explained 10%, 51%, and 62% of the variance in mood regulation, deficient self-regulation, and negative outcomes scores, respectively. These results support H5–H7.

## Discussion

The goals of the present study were to evaluate the psychometric properties of the Japanese version of the GPIUS2 and assess the adequacy of the theoretical model of generalized PIU for Japanese adults.

Regarding the psychometric evaluation, EFA was used to investigate the factor structure of the Japanese version of the GPIUS2. The results revealed that the model was slightly different from the original version of Caplan [18] in terms of item composition. Our model has a similar five-factor structure to the original version [18], although one item was classified as a "compulsive internet use" rather than a "cognitive preoccupation" item. Considering that the contents of item 12 could be categorized as either obsessive thoughts about internet use (i.e., cognitive preoccupation) or inadequate behavioural control (i.e., compulsive use), we classified it as a "cognitive preoccupation" item. This difference in item classification might be due to cultural differences in the nature of obsessive-compulsive symptoms between Japan and other countries while the results of the only study on the factor structure of obsessive-compulsive symptoms in Japan [31] were consistent with those of Western studies (for a quantitative review, see [32]). Anyhow, because cognitive preoccupation and compulsive use are forms of deficient self-regulation in the cognitive-behavioural model of Caplan [18], the difference between the original and the Japanese version is considered a minor issue.

Subsequent CFA to confirm the measurement model and assess its construct validity revealed that the Japanese model, which had first-order and second-order factors, was more valid than the other models, including Caplan's original model [18]. The results support the notion that the Japanese version of the GPIUS2 has a multifactorial structure suitable for assessing generalized PIU. In addition, the adequate fit of the factor model, as well as the acceptable internal consistency and significant correlations between GPIUS2 subscale scores and scores on questionnaires such as the IAT and RSES, indicate that the Japanese version of the GPIUS2 is useful for measuring generalized PIU in Japanese adults.

SEM was performed to verify the applicability of the cognitive-behavioural model proposed by Caplan [18] to the Japanese adult population. All four direct relationships observed in previous GPIUS2 validation studies [12, 18, 21, 22] were also observed in the current study. POSI was a positive predictor of internet use for mood regulation purposes, and the extent of deficient self-regulation. The use of the internet for mood regulation was a positive predictor of deficient self-regulation. Deficiency in self-regulation was a positive predictor of adverse

outcomes of internet use. In addition, the current study observed the three indirect relationships reported in previous studies [12, 18, 21, 22]. Mood regulation mediates the positive relationship between POSI and deficiency in self-regulation. Deficiency in self-regulation mediates the positive relationship between POSI and negative outcomes, and the positive relationship between mood regulation and negative outcomes. These relationships suggest that POSI, mood regulation, and deficient self-regulation play essential roles in generalized PIU and support the multidimensional generalized PIU model proposed by previous studies [9, 14–17].

The results of the current study are clinically relevant. The internet is widely used in East Asian countries, including Japan (https://data.worldbank.org/indicator/IT.NET.USER.ZS?end=2020&start=2020&view=map), and high prevalence rates of gaming disorder have been reported in the region (for a systematic review, see [33]). A specific type of gaming disorder (i.e., predominantly involving online games) [34] is associated with generalized PIU. Moreover, generalized PIU, including online gaming, often coexists with the "Hikikomori" (social withdrawal over a long period) phenomenon seen in Japan [35]. As well as generalized PIU, Hikikomori is influenced by multiple factors that partially overlap with generalized PIU [36], and has become a major social issue in Japan. The Japanese version of the GPIUS2 will allow for a multidimensional analysis of these related conditions, and could aid the development of evidence-based interventions by identifying non-adaptive cognitions and behaviours.

The present study had some limitations. First, the numbers of participants who completed the various questionnaire differed, which could have affected the results of the convergent validity analysis. However, the results of the correlation analysis were almost identical between the entire cohort and subgroups that completed each questionnaire (S1 Table). Thus, the convergent validity was sufficient. Second, this study included healthy adults. Most participants probably did not present with PIU, based on the IAT results. Future studies may need to confirm the results of the present study in populations with PIU or a high risk thereof, although the GPIUS2 had good sensitivity.

In summary, the current study demonstrated that the Japanese version of the GPIUS2, which enables evaluation of generalized PIU from a multidimensional perspective, has good psychometric properties. Moreover, the study provides further evidence of the utility of the cognitive-behavioural model of generalized PIU.

## Supporting information

**S1 Table. Partial correlation coefficients between the GPIUS2 scale scores and IAT scores, self-esteem, depression, and impulsivity with age as a covariate (listwise deletion method; $n$ = 128).**
(DOCX)

## Acknowledgments

The authors thank the study participants and Ms. Ikegami, who assisted with the operating procedures.

## Author Contributions

**Conceptualization:** Sayaka Yoshimura, Hironobu Fujiwara.

**Data curation:** Sayaka Yoshimura, Mami Shibata, Yusuke Kyuragi, Kei Kobayashi, Morio Aki, Hironobu Fujiwara.

**Formal analysis:** Sayaka Yoshimura.

**Funding acquisition:** Sayaka Yoshimura, Toshiya Murai, Hironobu Fujiwara.

**Investigation:** Sayaka Yoshimura, Mami Shibata.

**Methodology:** Sayaka Yoshimura.

**Project administration:** Hironobu Fujiwara.

**Supervision:** Toshiya Murai, Hironobu Fujiwara.

**Visualization:** Morio Aki.

**Writing – original draft:** Sayaka Yoshimura.

**Writing – review & editing:** Mami Shibata, Yusuke Kyuragi, Kei Kobayashi, Morio Aki, Toshiya Murai, Hironobu Fujiwara.

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
