## [Decision Letter · Decision Letter 0]

4 Jul 2022

PONE-D-22-15335The Japanese version of the Generalized Problematic Internet Use Scale 2 (GPIUS2): Psychometric Evaluation and Analysis of the Theoretical ModelPLOS ONE

Dear Dr. Sayaka Yoshimura

Thank you for submitting your manuscript to PLOS ONE. After careful consideration, we feel that it has merit but does not fully meet PLOS ONE’s publication criteria as it currently stands. Therefore, we invite you to submit a revised version of the manuscript that addresses the points raised during the review process.

We look forward to receiving your revised manuscript.

Kind regards,

Carmen Concerto

Academic Editor

PLOS ONE

Journal Requirements:

2. Please change "female” or "male" to "woman” or "man" as appropriate, when used as a noun (see for instance https://apastyle.apa.org/style-grammar-guidelines/bias-free-language/gender).

"The authors have no conflict of interest."

Reviewers' comments:

Reviewer's Responses to Questions

**Comments to the Author**

1. Is the manuscript technically sound, and do the data support the conclusions?

Reviewer #1: Yes

Reviewer #2: Yes

2. Has the statistical analysis been performed appropriately and rigorously? 

Reviewer #1: Yes

Reviewer #2: Yes

3. Have the authors made all data underlying the findings in their manuscript fully available?

Reviewer #1: No

Reviewer #2: No

4. Is the manuscript presented in an intelligible fashion and written in standard English?

Reviewer #1: Yes

Reviewer #2: Yes

5. Review Comments to the Author

Reviewer #1: The reviewed manuscript is a translation and validation of the Generalized Problematic Internet Use Scale 2 in the Japanese language. The manuscript is well written, and the statistical analyses are sounding. The following points should be addressed before the manuscript is accepted:

- Line 60: the authors generically mention the relevance in psychiatry of the PIU construct by mentioning the correlation between PIU and other psychiatric disorders. However, the systematic review that the authors mention is quite old, and new studies (original and reviews) have been produced on the topic and should be considered for citation. A few are listed below:

- Concerto, C., Rodolico, A., Avanzato, C., Fusar-Poli, L., Signorelli, M. S., Battaglia, F., & Aguglia, E. (2021). Autistic Traits and Attention-Deficit Hyperactivity Disorder Symptoms Predict the Severity of Internet Gaming Disorder in an Italian Adult Population. Brain sciences, 11(6), 774. https://doi.org/10.3390/brainsci11060774

- Ostinelli, E. G., Zangani, C., Giordano, B., Maestri, D., Gambini, O., D'Agostino, A., Furukawa, T. A., & Purgato, M. (2021). Depressive symptoms and depression in individuals with internet gaming disorder: A systematic review and meta-analysis. Journal of affective disorders, 284, 136–142. https://doi.org/10.1016/j.jad.2021.02.014

- Moretta, T., & Buodo, G. (2021). The Relationship Between Affective and Obsessive-Compulsive Symptoms in Internet Use Disorder. Frontiers in psychology, 12, 700518. https://doi.org/10.3389/fpsyg.2021.700518

- Line 122 and following: the authors mention that the patients were previously recruited in two studies, but no details are provided. The exact recruitment strategy must be reported, as also the setting. Moreover, the authors mention the “imaging study on schizophrenia”. Even if the subsequent sentences clarify that only patients not having psychiatric or neurologic conditions were considered, it is hard to understand the role of healthy subjects in a study such as the one the authors mentioned.

- Line 155: the authors should define IAT in the title of the paragraph

- Line 206: the statistical analysis paragraph focuses only on the validation procedure, but the authors adopted other statistical procedures, such as evaluating the normality of the distribution of the variables.

- Line 207: You mention the study hypotheses but have not mentioned any hypothesis before. The sentence should be rephrased, or the last sentences of the introduction (on objectives and aims) should better clarify the hypotheses tested.

- Line 243: the supplementary figure 1 is the pillar of most of this manuscript, and it is extensively discussed in the discussion section. Please consider adding it as in-text material.

- Line 266: you reported that almost all variables are non normally distributed as reported by the Shapiro-Wilk Statistic. In this case, it would be preferable to report median and interquartile ranges instead of means and SDs. Ranges are not sufficient.

- Line 341: Tabel 4 is a correlation matrix. The authors should consider applying the correction for multiple comparisons. Moreover, your significance cut-off should be reported in methods.

- Line 386: I could not understand the meaning of dispense in the line.

- Line 455: the sentence “the present study may not be generalizable to individuals with PIU” is a bit crude. The utility of a psychometric tool is also to have good sensitivity. Then, the authors could consider rephrasing the sentence.

Reviewer #2: Thank you for inviting me to review the paper entitled “The Japanese version of the Generalized Problematic Internet Use Scale 2 (GPIUS2): Psychometric Evaluation and Analysis of the Theoretical Model”. In this work, the psychometric properties of the GPIUS2 - a self-administered questionnaire evaluating problematic internet use (PIU) – were tested in a sample of 291 healthy Japanese adults.

The metodology of the study is strong and the paper is very well written and clear. The results are discussed in an adequate manner, also providing some clues for clinical practice and future research. I only have a few comments for the authors:

1) Introduction: In the first two paragraphs the authors provide a broad definition of PIU. Howwever, they do not specify when internet use is defined “problematic” and how PIU can be distinguished from “normal” internet use (i.e., duration in time, intensity?). Please, describe more in detail.

2) Age range of participants was 18-78. Please, reported also in the paragraph about "Participants" and specify that 18 years was the cut-off for excluding participants.

3) I guess that “healthy controls” is not completely appropriate, as they were not screened for medical conditions. Please, specify that that they were free ONLY from mental and neurological disorders.

6. PLOS authors have the option to publish the peer review history of their article (what does this mean?). If published, this will include your full peer review and any attached files.

Reviewer #1: No

Reviewer #2: **Yes: **Laura Fusar-Poli

---

## [Author Response · Author response to Decision Letter 0]

4 Aug 2022

Response to Journal requirements

Comment 1: Please ensure that your manuscript meets PLOS ONE's style requirements, including those for file naming.

Authors’ reply: We revised our manuscript to meet the style requirements.

Comment 2: Please change "female” or "male" to "woman” or "man" as appropriate, when used as a noun.

Authors’ reply: We changed “female” and “male” to “woman” and “man”, respectively, in the revised manuscript.

Comment 3: Please complete your Competing Interests on the online submission form to state any Competing Interests. If you have no competing interests, please state "The authors have declared that no competing interests exist.", as detailed online in our guide for authors. This information should be included in your cover letter; we will change the online submission form on your behalf.

Authors’ reply: Because the authors have declared that no competing interests exist, we would be grateful if you could change the form.

Comment 4: In your Data Availability statement, you have not specified where the minimal data set underlying the results described in your manuscript can be found. We will update your Data Availability statement to reflect the information you provide in your cover letter.

Authors’ reply: Please change the statement as follows; 

Data cannot be shared publicly for ethical reasons. This project has not been granted ethics to release the data publicly and we did not obtain the consent of participants to provide the datasets to third parties. However, researchers who meet the criteria for access to confidential data can contact the Ethics Committee of Kyoto University Graduate School and Faculty of Medicine at: ethcom@kuhp.kyoto-u.ac.jp; or the corresponding author at: yoshimura.sayaka.3x@kyoto-u.ac.jp.

Response to Reviewer #1

Comment 1: Line 60: the authors generically mention the relevance in psychiatry of the PIU construct by mentioning the correlation between PIU and other psychiatric disorders. However, the systematic review that the authors mention is quite old, and new studies (original and reviews) have been produced on the topic and should be considered for citation. A few are listed below:

- Concerto, C., Rodolico, A., Avanzato, C., Fusar-Poli, L., Signorelli, M. S., Battaglia, F., & Aguglia, E. (2021). Autistic Traits and Attention-Deficit Hyperactivity Disorder Symptoms Predict the Severity of Internet Gaming Disorder in an Italian Adult Population. Brain sciences, 11(6), 774. https://doi.org/10.3390/brainsci11060774

- Ostinelli, E. G., Zangani, C., Giordano, B., Maestri, D., Gambini, O., D'Agostino, A., Furukawa, T. A., & Purgato, M. (2021). Depressive symptoms and depression in individuals with internet gaming disorder: A systematic review and meta-analysis. Journal of affective disorders, 284, 136–142. https://doi.org/10.1016/j.jad.2021.02.014

- Moretta, T., & Buodo, G. (2021). The Relationship Between Affective and Obsessive-Compulsive Symptoms in Internet Use Disorder. Frontiers in psychology, 12, 700518. https://doi.org/10.3389/fpsyg.2021.700518

Authors’ reply: We thank Reviewer #1 for this helpful comment. We revised the Introduction and briefly mention new studies in the revised manuscript, including the ones listed above (Line 57-58).

Comment 2: Line 122 and following: the authors mention that the patients were previously recruited in two studies, but no details are provided. The exact recruitment strategy must be reported, as also the setting. Moreover, the authors mention the “imaging study on schizophrenia”. Even if the subsequent sentences clarify that only patients not having psychiatric or neurologic conditions were considered, it is hard to understand the role of healthy subjects in a study such as the one the authors mentioned.

Authors’ reply: The participants were enrolled in two research projects; we have better-described how we used their data in the Participants section of the revised manuscript (Line 129-133, 136-137).

Comment 3: Line 155: the authors should define IAT in the title of the paragraph.

Authors’ reply: We now spell out IAT in the title (Line 165).

Comment 4: Line 206: the statistical analysis paragraph focuses only on the validation procedure, but the authors adopted other statistical procedures, such as evaluating the normality of the distribution of the variables.

Authors’ reply: We revised the Data Analysis section and now mention the descriptive statistics and Shapiro–Wilk test (Line 217-218). Additionally, we described the results of the Shapiro–Wilk test in the Descriptive Statistics section of the Results (Line 273-274).

Comment 5: Line 207: You mention the study hypotheses but have not mentioned any hypothesis before. The sentence should be rephrased, or the last sentences of the introduction (on objectives and aims) should better clarify the hypotheses tested.

Authors’ reply: We added the hypotheses to the last paragraph of the Introduction (Line 111-121). In addition, we refer to the hypotheses in the Results section (Line 388-389, Line 396-397).

Comment 6: Line 243: the supplementary figure 1 is the pillar of most of this manuscript, and it is extensively discussed in the discussion section. Please consider adding it as in-text material.

Authors’ reply: We renamed Supplementary Figure 1 as Fig 2 and have added it to the revised manuscript as a key figure. In line with this, Figs 2 and 3 are now Figs 3 and 4, respectively.

Comment 7: Line 266: you reported that almost all variables are non normally distributed as reported by the Shapiro-Wilk Statistic. In this case, it would be preferable to report median and interquartile ranges instead of means and SDs. Ranges are not sufficient.

Authors’ reply: We report median and interquartile ranges for the variables, instead of means and standard deviation, in the revised manuscript (Line 30-31, Line 128-129, Line 179-180, Line 190-191, Line 202-203, Line 213-214, Table 1).

Comment 8: Line 341: Table 4 is a correlation matrix. The authors should consider applying the correction for multiple comparisons. Moreover, your significance cut-off should be reported in methods.

Authors’ reply: We revised Table 4 to include the results of multiple comparisons.

Comment 9: Line 386: I could not understand the meaning of dispense in the line.

Authors’ reply: We agree that the use of “dispense” was confusing. We have revised the text to improve readability (Line 407).

Comment 10: Line 455: the sentence “the present study may not be generalizable to individuals with PIU” is a bit crude. The utility of a psychometric tool is also to have good sensitivity. Then, the authors could consider rephrasing the sentence.

Authors’ reply: We agree and have revised the text; please see Line 474-476, of the revised manuscript.

Response to Reviewer #2

Comment 1: Introduction: In the first two paragraphs the authors provide a broad definition of PIU. However, they do not specify when internet use is defined “problematic” and how PIU can be distinguished from “normal” internet use (i.e., duration in time, intensity?). Please, describe more in detail.

Authors’ reply: We have provided additional explanations about PIU in the revised Introduction (Line 51-54).

Comment 2: Age range of participants was 18-78. Please, reported also in the paragraph about "Participants" and specify that 18 years was the cut-off for excluding participants.

Authors’ reply: The target age of the two studies in which the participants took part as controls was ≥ 18 years, as described in the revised version (Line 129).

Comment 3: I guess that “healthy controls” is not completely appropriate, as they were not screened for medical conditions. Please, specify that that they were free ONLY from mental and neurological disorders.

Authors’ reply: We have specified this in the revised manuscript (Line 131-133).

---

## [Editor Report · Decision Letter 1]

8 Aug 2022

PONE-D-22-15335R1The Japanese version of the generalized problematic internet use scale 2 (GPIUS2): Psychometric evaluation and analysis of the theoretical ModelPLOS ONE

Dear Dr. Sayaka Yoshomura,

Thank you for submitting your manuscript to PLOS ONE. After careful consideration, we feel that it has merit but does not fully meet PLOS ONE’s publication criteria as it currently stands. Therefore, we invite you to submit a revised version of the manuscript that addresses the points raised during the review process.

ACADEMIC EDITOR: 

Please consider minor typo errors. 

Please remove the suggested reference n 4 (line 499)

A rebuttal letter that responds to each point raised by the academic editor. A marked-up copy of your manuscript that highlights changes made to the original version. You should upload this as a separate file labeled 'Revised Manuscript with Track Changes'.An unmarked version of your revised paper without tracked changes. You should upload this as a separate file labeled 'Manuscript'.If applicable, we recommend that you deposit your laboratory protocols in protocols.io to enhance the reproducibility of your results. Protocols.io assigns your protocol its own identifier (DOI) so that it can be cited independently in the future. For instructions see: https://journals.plos.org/plosone/s/submission-guidelines#loc-laboratory-protocols. Additionally, PLOS ONE offers an option for publishing peer-reviewed Lab Protocol articles, which describe protocols hosted on protocols.io. Read more information on sharing protocols at https://plos.org/protocols?utm_medium=editorial-email&utm_source=authorletters&utm_campaign=protocols.

We look forward to receiving your revised manuscript.

Kind regards,

Carmen Concerto

Academic Editor

PLOS ONE
---

## [Author Response · Author response to Decision Letter 1]

12 Aug 2022

Response to academic editor

Comment 1: Please consider minor typo errors. 

Authors’ reply: We have corrected typing errors in the new manuscript.

Comment 2: Please remove the suggested reference n 4 (line 499)

Authors’ reply: We removed reference number 4 in the old manuscript. Additionally, subsequent literature numbers have been changed.

Response to Journal requirements

Comment 1: Please review your reference list to ensure that it is complete and correct. If you have cited papers that have been retracted, please include the rationale for doing so in the manuscript text, or remove these references and replace them with relevant current references. Any changes to the reference list should be mentioned in the rebuttal letter that accompanies your revised manuscript. If you need to cite a retracted article, indicate the article’s retracted status in the References list and also include a citation and full reference for the retraction notice.

Authors’ reply: After reviewing and revising our manuscript (reference number 4 in the old manuscript was removed), we have confirmed that our reference list is complete and correct.

---

## [Editor Report · Decision Letter 2]

18 Aug 2022

The Japanese version of the generalized problematic internet use scale 2 (GPIUS2): Psychometric evaluation and analysis of the theoretical Model

PONE-D-22-15335R2

Dear Dr. Yoshimura

We’re pleased to inform you that your manuscript has been judged scientifically suitable for publication and will be formally accepted for publication once it meets all outstanding technical requirements.

Kind regards,

Carmen Concerto

Academic Editor

PLOS ONE
---

## [Editor Report · Acceptance letter]

22 Aug 2022

PONE-D-22-15335R2 

The Japanese version of the generalized problematic internet use scale 2 (GPIUS2): Psychometric evaluation and analysis of the theoretical Model 

Dear Dr. Yoshimura:

I'm pleased to inform you that your manuscript has been deemed suitable for publication in PLOS ONE. Congratulations! Your manuscript is now with our production department. 

Kind regards, 

on behalf of

Dr. Carmen Concerto 

Academic Editor

PLOS ONE